# Time Series Remote Sensing Image Classification with a Data-Driven Active Deep Learning Approach

**DOI:** 10.3390/s25061718

**Published:** 2025-03-10

**Authors:** Gaoliang Xie, Peng Liu, Zugang Chen, Lajiao Chen, Yan Ma, Lingjun Zhao

**Affiliations:** 1Aerospace Information Research Institute, Chinese Academy of Sciences, Beijing 100094, China; xiegaoliang22@mails.ucas.ac.cn (G.X.); chenzg@aircas.ac.cn (Z.C.); chenli@radi.ac.cn (L.C.); mayan@radi.ac.cn (Y.M.); zhaolj@radi.ac.cn (L.Z.); 2School of Electronic, Electrical and Communication Engineering, University of Chinese Academy of Sciences, Beijing 100049, China

**Keywords:** satellite image time series, labeling efforts, land use/land cover (LULC) mapping

## Abstract

Recently, Time Series Remote Sensing Images (TSRSIs) have been proven to be a significant resource for land use/land cover (LULC) mapping. Deep learning methods perform well in managing and processing temporal dependencies and have shown remarkable advancements within this domain. Although deep learning methods have exhibited outstanding performance in classifying TSRSIs, they rely on enough labeled time series samples for effective training. Labeling data with a wide geographical range and a long time span is highly time-consuming and labor-intensive. Active learning (AL) is a promising method of selecting the most informative data for labeling to save human labeling efforts. It has been widely applied in the remote sensing community, except for the classification of TSRSIs. The main challenge of AL in TSRSI classification is dealing with the internal temporal dependencies within TSRSIs and evaluating the informativeness of unlabeled time series data. In this paper, we propose a data-driven active deep learning framework for TSRSI classification to address the problem of limited labeled time series samples. First, a temporal classifier for TSRSI classification tasks is designed. Next, we propose an effective active learning method to select informative time series samples for labeling, which considers representativeness and uncertainty. For representativeness, we use the K-shape method to cluster time series data. For uncertainty, we construct an auxiliary deep network to evaluate the uncertainty of unlabeled data. The features with rich temporal information in the classifier’s middle-hidden layers will be fed into the auxiliary deep network. Then, we define a new loss function with the aim of improving the deep model’s performance. Finally, the proposed method in this paper was verified on two TSRSI datasets. The results demonstrate a significant advantage of our method over other approaches to TSRSI. On the MUDS dataset, when the initial number of samples was 100 after our method selected and labeled 2000 samples, an accuracy improvement of 4.92% was achieved. On the DynamicEarthNet dataset, when the initial number of samples was 1000 after our method selected and labeled 2000 samples, an accuracy improvement of 7.81% was attained. On the PASTIS dataset, when the initial number of samples was 1000 after our method selected and labeled 2000 samples, an accuracy improvement of 4.89% was achieved. Our code is available in Data Availability Statement.

## 1. Introduction

With the remarkable progress and rapid advancement of remote sensing satellite technology, there has been a significant boost in its data acquisition capabilities, consequently leading to the accumulation of a vast amount of time series remote sensing data. Remote sensing data have numerous applications, such as semantic segmentation [1,2,3,4], change detection [5,6,7,8], and so on. Time series remote sensing data have presented even greater application value. Thanks to the potential of their informative temporal information, they have been proven to be an important resource for land use/land cover mapping. Unlike the spectral band dimension, the time series represents a sequence with strong correlations. It is an unfolding in the time dimension that indicates the past, the present, and the future, and it contains abundant temporal information waiting to be explored.

Deep learning methods are good at handling complex temporal features and have become a popular research approach for Time Series Remote Sensing Image (TSRSI) classification [9,10,11,12,13,14] in recent years. Deep learning models such as Convolutional Neural Network (CNN) [15], Recurrent Neural Network (RNN) [16], and Long Short-Term Memory Network (LSTM) [17] have been successfully applied to TSRSI data and have demonstrated powerful classification performances [13,18]. Moreover, some scholars have incorporated attention mechanisms into deep networks to accurately capture temporal dependencies and achieve higher classification accuracy [9,18,19,20]. The deep learning models mentioned above for TSRSI classification can be divided into two types, according to the differences in their network output patterns: the many-to-many model and the many-to-one model. The many-to-many model means that the input is a sequence, and the output is also a sequence. This type of model is usually applied in scenarios where classification at each time point is required, such as the TSRSI classification tasks [14,17,21,22]. The many-to-one model means that the input is a sequence, and the output is a single value. It can classify an entire time series and is usually used in tasks such as text classification [23,24] and sentiment analysis [25,26].

There are relatively abundant time series classification models based on deep learning, and they have more robust capabilities for processing time series data. However, deep-learning-based models require sufficient labeled samples for training to achieve excellent classification performance. The shortage of over-dependence on labeled samples has already become a bottleneck for the practical application of these algorithms. It has always been a rather challenging task [27,28,29] to obtain sufficient training samples for supervised classifiers, especially for classifiers of the many-to-many type. There are two reasons why obtaining training samples for the many-to-many model is difficult. The first reason is that, in the spatial dimension, the geographical distribution of the data used for the many-to-many model is usually quite extensive. The second reason is that, in the time dimension, the data used for the many-to-many model cover a long time span. Therefore, it is very time-consuming and labor-intensive to label data from many locations at many different times. Field investigation and manual interpretation are two traditional methods of obtaining training samples in the field of remote sensing. Since time series data cover a wide geographical range and a long time span, the field investigation approach appears rather inefficient. The method of manual interpretation avoids the cost of conducting investigations in real scenes. However, since the data need to be labeled frame by frame and region by region, there is still the problem of low labeling efficiency. Moreover, due to the redundancy of remote sensing data, it is also difficult for this method to avoid the labor cost spent on unimportant data. Research on labeling strategies is urgent. They help us find key samples from many unlabeled TSRSIs, and significant labeling costs can be saved.

Active learning (AL) belongs to the study of labeling schemes. It is a promising way of selecting the most informative data for labeling to save on human labeling efforts. For example, satellite image time series data classification for crop mapping can offer valuable information for multiple agricultural applications, such as crop monitoring, yield forecasting, and crop inventory. SaeidNiazmardi [30] proposed an AL method to address the unavailability of adequate high-quality time series samples for training the classifier in the classification task. The AL framework for classification usually consists of a classifier and a selector. Constructing a good selector is one of the key problems in AL. The selector will select the most informative samples from the unlabeled dataset according to the classifier’s feedback, the data distribution, or other information. We only label these most critical samples and then add them to the training set, which enables us to achieve greater improvement in the performance of the classifier while labeling fewer samples.

Although many different types of AL methods have been applied to practical tasks in the field of remote sensing [31,32,33,34], most of these methods are aimed at the classification tasks of hyperspectral remote sensing images. However, research on AL methods for the classification of TSRSIs [30] is lacking. Time series remote sensing data have not only a spectral dimension, but also periodic or non-periodic time-dependent relationships. Therefore, it is not easy to directly use the existing metric methods for hyperspectral data to evaluate the informativeness of time series remote sensing data. There are many AL methods for time series classification in the field of computer science, but they cannot be directly applied to the field of remote sensing, mainly due to two limitations. The first limitation lies in the difference in the output modes of the tasks. Most AL methods [35,36,37,38,39] in the field of computer science are aimed at many-to-one type tasks, while our TSRSI classification tasks are of the many-to-many type. The second limitation is that there is a significant difference between the TSRSI data and the data, such as text and audio data, to which these methods are applied. Compared with them, TSRSI data have a higher dimension and greater redundancy.

Furthermore, the existing AL methods for hyperspectral image data are difficult to apply directly to the classification of TSRSIs. For AL for TSRSI data, it should be designed to utilize the temporal dependencies in time series remote sensing data; then, it could accurately judge the information richness of time series samples and effectively select critical time series samples for the temporal classifiers. Although [30] proposed an AL method for TSRSIs, this method is a model-driven one and lacks adaptability to different types of data [40]. We believe that the data-driven modes are more promising, since they can automatically learn features and patterns from a large amount of data, have strong adaptability to different types of data, and have high generalization performance.

In this paper, we propose a novel active deep learning method for TSRSI classification, which addresses the problem of limited labeled time series samples. This method selects samples based on both representativeness and uncertainty. For representativeness, we address the metric by using the K-shape method to cluster time series samples. For uncertainty, we design a deep model called a time series loss prediction module (TLPM) that is equipped with a self-attention mechanism. This deep model, based on the idea of data-driven models, can automatically learn features and patterns from a large amount of data and capture deep temporal dependencies. To verify the effectiveness of the proposed method, we conduct experiments on two TSRSI datasets. The experimental results demonstrate the superior performances of the proposed method compared with other AL methods.

The main innovations of this study can be summarized as follows:We proposed an AL framework based on representativeness and uncertainty for TSRSI classification. This framework selects key time series samples with richer information for annotation to address the problem of limited labeled time series samples and to save on labeling efforts.We have designed a selector named time series loss prediction module in the AL framework. The selector is a data-driven one that can automatically learn features and patterns from a large amount of data and capture deep temporal dependencies, which gives it higher generalization performance compared to model-driven methods.Our experiments were conducted on three datasets, MUDS [41], DynamicEarthNet [42], and PASTIS [43]. The data in these three datasets cover a wide range and are strongly geographically representative. The proposed method performs excellently on these three datasets. Experiments have demonstrated that our method can be applied to a variety of geographical environments.

## 2. Related Work

In the remote sensing community, labeling the data with a wide geographical range and a long time span is very time-consuming and labor-intensive if field investigation or manual interpretation is adopted. Active learning (AL) is a promising method used to select the most informative data for labeling to save on human labeling efforts. According to whether the selector is model-driven or data-driven, AL methods can be divided into two types: model-driven and data-driven [40].

### 2.1. Model-Driven AL Methods

Model-driven AL methods usually require us to design selectors based on handcrafted features or metrics to select the samples with the richest information. Model-driven AL methods can be roughly divided into three groups, as follows.
(1)Model-Driven AL with Uncertainty: Uncertainty-based AL methods hold that the most ambiguous samples with high uncertainty for the present model are the most effective in improving its accuracy if they are added to the training set. Some uncertainty-based AL methods aim to find samples located on the classification boundaries. For instance, ref. [44] proposes a breaking-ties AL method for a Bayesian approach. Ref. [45] introduces an entropy-based AL method to Logistic Regression. Ref. [46] presents a spatial batch-mode AL method based on margin distance to select uncertainty samples for a binary SVM classifier. A Maximum Confidence Uncertainty is applied in [47], which can find highly informative samples and automatically balance the training distribution.(2)Model-Driven AL with Representativeness: Since the unlabeled samples are always redundant, representativeness-based AL methods demonstrate that if representative samples are selected, the training set can be enriched. Many AL studies [48,49] regard representativeness as an important selection criterion when selecting samples. For example, a K-center-greedy method is introduced [50] into Core-Set to choose representative samples. Additionally, Liu et al. [31] employ sparse representation by dictionary learning to seek representative samples. These methods can help the target classifier grasp the data distribution.(3)Model-Driven AL with Model Influences: Model-driven AL methods with model influences select samples that have a significant impact on the model parameters of the target model. The model influences can be measured by Fisher Information [51,52,53], Expected-Gradient-Length (EGL), etc. The paper [54] applies Fisher information to analyze the objectives in the context of AL asymptotically, aiming to provide theoretical support and insights for optimizing the performance of AL algorithms. The first EGL strategy was proposed in [55] to select samples with a high magnitude gradient.

### 2.2. Data-Driven AL Methods

The selectors in data-driven AL methods are usually deep models and are closely related to deep learning. These methods are often associated with uncertainty, reinforcement learning, and data augmentation.
(1)Data-Driven AL Methods with Uncertainty: The difference from the model-driven uncertainty-based AL methods is that uncertainty in data-driven AL methods is usually measured by automatic feature learning, such as hidden layers. Loss learning [56], discriminate learning [57], and adversarial learning [58] are all data-driven AL methods with uncertainty. Among them, [56] proposes a novel AL method that attaches a small parametric module called the loss prediction module to a target network to predict the target losses of unlabeled samples. Here, the target loss is an embodiment of uncertainty. An adversarial uncertainty-based AL method is proposed in [59] to query valuable samples. VAAL [60] learns a latent space using a variational autoencoder (VAE) and an adversarial network trained to discriminate between unlabeled and labeled data, which combines discriminate learning and adversarial learning to select uncertainty samples.(2)Data-Driven AL Methods with Reinforcement Learning: AL methods with reinforcement learning empower software-defined agents to actively explore and figure out the most optimal actions within a virtual environment. This streamlines the sample selection process for AL and injects a dynamic, self-learning mechanism. In [61], a reinforced pool-based deep AL approach to select informative samples for annotation is proposed, which can dynamically select valuable samples for annotation and adaptively optimize classification strategies.(3)Data-Driven AL Methods with Data Augmentation: Unlike other methods, which select real samples from the unlabeled dataset, data augmentation-based AL methods aim to generate some samples with rich information. GAAL [62] introduces the GANs into AL, which integrates the ideas of data augmentation and uncertainty to generate and select valuable samples, respectively. The BGAL [63] first performs the operation of selecting samples, then generates samples with rich information and adds them to the candidate dataset.

AL methods have been widely applied in the remote sensing community except for TSRSI classification. The main challenge of AL in TSRSI classification is dealing with the internal temporal dependencies within TSRSIs and evaluating the informativeness of unlabeled time series data. In this paper, we propose a novel active deep learning method for TSRSI classification which addresses the problem of limited labeled time series samples. The proposed method considers the temporal dependencies of TSRSIs in defining their informativeness, which is measured by representativeness and uncertainty.

## 3. Methods

The AL method proposed in this paper consists of three components: a clustering module, a temporal classifier, and the TLPM. The clustering module is the K-Shape method to provide support for selecting temporal samples from a perspective of sample representativeness. The temporal classifier works in TSRSI classification. The TLPM predicts the loss of unlabeled temporal samples to evaluate the uncertainty of each time series instance. The flowchart of the proposed method is displayed in Figure 1. The process of sample selection comprises two main phases: training and inference, which are performed in a loop. (1) Before the loop starts, initial temporal samples are randomly sampled from a TSRSI dataset.

During the training phase, (2) samples in the training set are input into the temporal classifier model for training. (3) Simultaneously, the features extracted from the encoding section of the classifier are fed into the TLPM. The target loss from the temporal classifier is utilized as a label for training the TLPM. During the inference phase, (4,5) unlabeled data samples are first clustered, and the cluster labels are recorded. (6) The next step is to input these samples into the temporal classifier. (7) Then, the intermediate features are input into the TLPM. (8,9) After that, we select time series samples with the highest loss in each cluster to label them and add them to the train set.

### 3.1. LSTM-Based Temporal Classifier

TSRSI data contain rich temporal information which can be fully utilized for TSRSI classification by a temporal classifier based on deep learning. As shown in the upper right of Figure 1, our network is a pure temporal version designed based on [17], which is based on the U-Net framework with an embedded LSTM layer. This classifier was chosen because it contains convolutional layers and an LSTM module, making it somewhat representative of numerous temporal classifiers. The convolutional layers in U-net can capture the temporal relationships of neighbor pixels over time. Meanwhile, the LSTM in this classifier captures deep time dependencies. It should be noted that this classifier’s input and output mode is many-to-many. In other words, this classifier will predict a land cover category for the data at each time point, which aligns with the goal of our TSRSI classification task. We define our temporal classifier as $F_{c}$.

So far, this paper has introduced a temporal classifier that works on TSRSI classification. However, it needs a large or well-selected training set. The reason for this is that a training set with a small data size is inadequate to support the classifier in accurately identifying the decision boundary between classes. Therefore, we propose an AL method that considers representativeness and uncertainty to select informative samples to form a well-selected training set.

### 3.2. Select Samples Based on Representativeness and Uncertainty

We commence with a crafted illustration highlighting the significance of selecting samples based on both representativeness and uncertainty in AL. Figure 2a shows the data selected by an approach based on uncertainty. The method based on uncertainty can find the decision boundary. But it is more likely to select similar samples. Figure 2b shows the data selected based on both representativeness and uncertainty. Evidently, the approach integrating both representativeness and uncertainty proves more effective in delineating the precise decision boundary compared to relying solely on uncertainty.

In terms of representativeness, selecting unlabeled data from different clusters can help find more representative samples. We choose the K-Shape method [64] to divide unlabeled time series data into *K* clusters because of its relatively small computational complexity and focus on the shape of the time series. In terms of uncertainty, the greater the classification loss of a sample, the greater the uncertainty of the sample. Therefore, we propose the TLPM to predict the loss of the sample. The input part of the TLPM comes from the output features of different levels of the classifier in the encoding stage. Each input feature map will enter this submodule and undergo a series of transformations, which are defined here as $S_{m}$. Therefore, this network module $F_{u}\left(e_{m},\ldots ,e_{1}\right)$ can be defined as(1)$$F_{u}\left(e_{m},\ldots ,e_{1}\right)=FC\left(S_{m}\left(e_{m}\right),\ldots ,S_{1}\left(e_{1}\right)\right)$$
$e_{m}$ represents the m-th output feature of the encoder, while FC(·) stands for fully connected transformation. Unlike the network of [56], the $S_{m}$ here does not only include the GAP layer [65] and fully connected (FC) layer; we have made modifications to it. As shown in Figure 3, $S_{m}$ consists of three parts. The first part involves a self-attention mechanism, which relates different positions of a single time sequence to compute a representation of the time sequence [66]. It helps the module focus on the most critical time points when predicting the loss in time series data. The remaining two parts consist of the GAP and FC layers, respectively.

Currently, we have not defined a specific loss function. In the next section, we will explain how to design the loss function and the method of training the network.

### 3.3. Loss Function of Our Network Architecture

We defined a temporal classifier for TSRSI classification in Section 3.1 and the TLPM in Section 3.2. Next, to enable the TLPM to accurately evaluate the uncertainty of unlabeled data, we will define the loss function.

By inputting samples with labels into the classifier, we can obtain a loss value by comparing the true labels with the predicted ones. However, since unlabeled samples do not have corresponding labels, we cannot directly obtain the loss values for unlabeled data. Instead, we predict the loss value of a time series instance through the TLPM. First, for the classifier, we define the output probability of x as $\hat{z}=F_{c}\left(x\right)$, so the loss value for a single sample *i* is(2)$$l_{c}^{i}=L_{c}\left(z,\hat{z}\right)$$
$L_{c}$ represents the loss function of the classifier, where z is the true label of the sample and $z=[z_{1},\cdots ,z_{T}]$. For the TLPM $F_{u}$, its predicted loss is denoted as $l_{u}^{i}$, and $l_{c}^{i}$ is the corresponding actual loss value. Therefore, the loss of this module is defined as $L_{u}\left(l_{c}^{i},l_{u}^{i}\right)$. The total loss is defined as(3)$$L=L_{c}\left(z,\hat{z}\right)+\lambda L_{u}\left(l_{c}^{i},l_{u}^{i}\right)$$
λ is an adjustable parameter used to represent a scaling constant in the loss optimization process. Specifically, we use cross-entropy loss as loss function for the classifier. When a batch contains *N* temporal samples, the cross-entropy loss can be expressed as(4)$$L_{c}=-\frac{1}{NT}\sum_{i=1}^{N}\sum_{t=1}^{T}z_{t}^{i}\log\left({\hat{z}}_{t}^{i}\right)$$
Defining the loss function for the TLPM is an important issue. During network training, while the classifier’s accuracy improves, its loss almost always decreases. It would be extremely challenging to train the TLPM using loss labels with large-scale variations. However, during network training, the relative loss values between temporal samples exhibit smaller variations. In other words, if we can compare loss functions between pairs of samples, we can discard the overall scale of loss. We designate a batch of trained data *N*, which is an even number. Thus, this batch contains N/2 pairs of temporal sample data, denoted as $x_{p}=\left\{x_{i},x_{j}\right\}$. The samples are paired in a head-to-tail manner. If the serial numbers of this batch of samples are 1,2,3…N, then the relationship between *i* and *j* here is i+j=N+1. We can then update the parameters of the TLPM by considering the difference between a pair of loss predictions, where the loss function is defined as(5)$$L_{u}\left(l^{p},\;\hat{l^{p}}\right)=\max\left(0,-sign\left(l_{c}^{i}-l_{c}^{j}\right)\cdot \left(l_{u}^{i}-l_{u}^{j}\right)+\psi \right)$$
Here, the symbol ψ represents a predefined positive number, $l^{p}$ represents a pair of temporal samples (i,j), and sign(·) represents the sign function. For example, when $l_{c}^{i}>l_{c}^{j}$ with $-\left(l_{u}^{i}-l_{u}^{j}\right)+\psi <0$, the loss function value is 0, providing no feedback to the network. Feedback from the loss values is received by the module only when $l_{c}^{i}$ > $l_{c}^{j}$ with $-\left(l_{u}^{i}-l_{u}^{j}\right)+\psi >0$, thereby increasing $l_{c}^{i}$ and decreasing $l_{c}^{j}$.The same applies when $l_{c}^{i}<l_{c}^{j}$. When the network trains a batch of data η with a size of *N*, the final loss function for the classifier and the TLPM is represented as(6)$$\frac{1}{N}\sum_{(x,z)\in \eta }L_{c}\left(z,\hat{z}\right)+\lambda \frac{2}{N}\sum_{(x_{p},z_{p})\in \eta }L_{u}\left(l^{p},\;\hat{l^{p}}\right)$$
During training, the losses of both the classifier and the TLPM are simultaneously reduced.

At present, we have completed the introduction of the Methods section. The algorithm implementation process of this method can be seen in Algorithm 1.
**Algorithm 1:** Algorithm for selecting time series samples
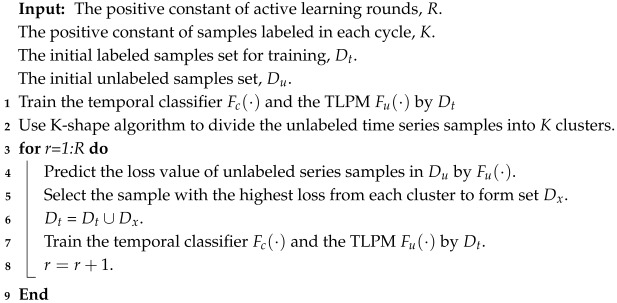


## 4. Experiments and Results

### 4.1. Dataset Details

To validate the proposed method, we conducted experiments on three datasets: the multi-temporal urban development spacenet dataset (MUDS) [41], the daily multi-spectral satellite dataset named DynamicEarthNet [42], and Panoptic Agricultural Satellite Time Series (PASTIS) [43]. The three datasets were chosen to conduct the experiments for two reasons: (a) they cover a wide geographical range, and the data show geographical diversity, which poses a strong challenge to a time series classifier and even an AL framework; (b) they have a large time span and contain a sufficient number of time points, demonstrating apparent seasonality or other temporal dependencies, which are somewhat representative in TSRSI data. These three datasets in detail are introduced in the following:

(1) MUDS: The MUDS dataset was captured in an urban scene in the city of Planet Labs’ Dove constellation between 2017 and 2020. Its image area exceeds 40,000 square kilometers and covers over 100 unique geographical locations. Each location includes one image per month, for a total of about 24 or 25 images. Each image frame has a size of 1024 × 1024 pixels and four spectral bands, with a spatial resolution of 4 m. The sample image of MUDS and their ground-truth classification map are shown in Figure 4. This dataset has two categories: built-up areas and non-built-up areas. The specific details can be seen in Table 1.

(2) DynamicEarthNet: The second dataset is the DynamicEarthNet dataset, which was acquired in Planet Labs from January 2018 to December 2019. It contains 75 areas of interest (AOIs) around the world. Each area of interest includes 24 images at different time points. Each image has a size of 1024 × 1024 pixels and four spectral bands, with a spatial resolution of 3 m. The sample image of DynamicearthNet and their ground-truth classification map are shown in Figure 5. This DynamicEarthNet dataset is cloud-free and contains pixel-wise monthly semantic segmentation labels of seven LULC classes, namely soil, agriculture, water, impervious surface, forest and other vegetation, snow, and wetland. The specific details can be seen in Table 1.

(3) PASTIS: The third dataset is the PASTIS dataset, which was captured by the Sentinel-2 satellite from September 2018 to November 2019 and taken from four different regions of France with diverse climates and crop distributions. It contains 2433 sequences of multi-spectral images of shape 10 × 128 × 128, with a spatial resolution of 10 m, and close to 28% of images have at least partial cloud cover. The sample image of PASTIS and its ground-truth classification map are shown in Figure 6. This PASTIS dataset contains pixel-wise semantic segmentation labels of 18 crop types plus a background class. The specific details can be seen in Table 1.

### 4.2. Competitive Approaches

The methods proposed in this paper are compared with other methods on the two datasets, including Random, Entropy, Variational Adversarial Active Learning (VAAL) [60], Margin, Core-Set, and Nearest Neighbor-based Active Learning (NNAL) [38]. (1) Random will randomly select from the unlabeled dataset and is often used as a baseline. (2) Entropy will evaluate the entropy of unlabeled samples as their uncertainty and samples with high entropy values will be selected. (3) VAAL learns a latent space of unlabeled and labeled data using a variational autoencoder (VAE) and a discriminator trained to discriminate between unlabeled and labeled data. It selects the samples that the discriminator is most uncertain about. (4) Margin is an uncertainty-based AL method and selects samples with a small margin that is defined as the difference between the probabilities of the two most likely classes. (5) Core-Set aims to find samples that can approximate the data distribution of the original dataset to a certain extent. In this paper, we implemented this method using the k-center-greedy algorithm. (6) NNAL uses highly local information to measure the uncertainty and utility of time series samples, which has demonstrated its effectiveness in many-to-one time series classification.

### 4.3. Experimental Settings and Implementation Details

The experiments were run on an Intel(R) Xeon(R) Silver 4210 2.20-GHz CPU and an NVIDIA GeForce RTX 4090 GPU from Beijing, China. We used code libraries such as NumPy (version 1.26.0), PyTorch (version 2.1.1), and scikit-learn in the Python (version 3.10.13) language to use the traditional methods. We designed a one-dimensional temporal classifier based on the modification of BANet [17], which contains convolutional layers and an LSTM module. To prevent the problem of decreased accuracy in temporal classifiers caused by overfitting, during the training of the time series classifier for 200 epochs, the network parameters that yield the highest accuracy on the validation set are chosen as the final parameters for the temporal classifier. The temporal classifier in this state is then used to test the accuracy of the final test set. When the GPU video memory is large enough, the larger the batch size, the faster the training speed. Therefore, a large batch size of 4096 was used. Parameters λ and ζ were set to 1, and the fully connected layer had 64 neurons.

To reduce the computational cost, we first conducted a uniform sampling operation on the original images in each dataset before running the AL method. Meanwhile, to avoid losing key information about the images, we chose four as the step size for the uniform sampling. Each approach was executed five times on each dataset. Each time, it was run on a randomly partitioned dataset with different random number seeds, which was divided into the training set, candidate set, validation set, and test set. The average of the results obtained by running each method five times was taken as the final accuracy on each dataset.

### 4.4. The Training Settings on Datasets

After the preliminary uniform sampling, we continue to divide the dataset randomly according to the proportion. For the MUDS dataset, we filtered out the data without cloud cover. Then we take 70% of the data as the test set and 10% of the data as the validation set, and from the remaining 20% of the data, we select 100 or 1000 samples randomly as the initial training set, with the rest of the data used as the candidate set. For the DynamicEarthNet dataset, the division ratio of the test set to the validation set is the same as that of the MUDS dataset. Because the DynamicEarthNet dataset is a multi-class dataset, more training samples are needed. We select 1000 or 10,000 samples randomly from the remaining 20% of the data as the initial training set, with the rest of the data used as the candidate set. For the PASTIS dataset, the division ratio of the test set and the validation set is also the same as that of the MUDS dataset. We select 1000 or 5000 samples randomly from the remaining 20% of the data as the initial training set, with the rest of the data as the candidate set.

When we regard the initial training set and the candidate set as an overall training set to conduct experiments, we achieved 84.61% in OA on the MUDS dataset, 84.78% in OA on the DynamicEarthNet dataset, and 75.02% in OA on the PASTIS dataset. Next, the experiment on MUDS is divided into two groups. In the first group, the initial training set size is set to 100, and the sample selection interval is also set to 500 samples. In the second group, the initial training set size is set to 1000, with a sample selection interval of 1500. Similarly, the experiments on DynamicEarthNet are divided into two groups. In the first group, the initial training set size is set to 1000, and the sample selection increment is set to 1000 samples. In the second group, the initial training set size is set to 10,000, and the sample selection increment is set to 1500 samples. The experiments on PASTIS are also divided into two groups like the DynamicEarthNet dataset. But in the second group, the initial training set size is set to 5000, and the sample selection increment is set to 1500 samples. The number of trainable samples in the first group of each dataset was relatively scarce. In the second group of each dataset, although the trainable samples were relatively abundant, the performance of the classifier did not meet the expected standards. The reason for choosing these two parameter sets in each dataset is to verify the effectiveness of our proposed method under different sample quantities.

The temporal classifier’s initial learning rate is set to 0.01, and during training, the Adam optimizer adaptively adjusts it. For the K-shape clustering method, we set the number of clusters to the corresponding selection interval. The TLPM’s initial learning rate is also set to 0.01. After 90 epochs, the learning rate decreases to 0.001. During each training process, gradient propagation from the TLPM to the temporal classifier stops after the 120th epoch to ensure the effective training of the time series classifier.

### 4.5. Effectiveness Verification

Our main experiments are shown in Figure 7 and Table 2. Random performs well when the data volume is small. This is because, with a smaller data volume, the classifier has a limited understanding of the sample distribution, and randomly selected samples can provide more information to the temporal classifier. As the number of samples increases, the temporal classifier gains more information about data distribution. Therefore, model-driven methods with uncertainty, like Entropy and Margin, both demonstrated certain effectiveness when there were sufficient samples. The Core-Set method that does not incorporate uncertainty performs poorly. NNAL performs moderately in the many-to-many time series classification tasks. This is because this method considers uncertainty and the distances between time series samples while ignoring the internal information correlations within the time series. The VAAL method performs worse because it is challenging for the variational auto-encoder to encode and decode time samples with temporal dependencies. Our method has learned correlation relationships within the time series and selects informative samples with uncertainty and representativeness. Therefore, our method has demonstrated its advantages under a variety of different sample quantities in the three datasets. The proposed method not only selects information-rich samples, but also takes into account the representativeness of the samples. Therefore, it can improve the classification performance for multiple categories.

In addition, for the MUDS dataset with a relatively small number of categories, the precision change curves oscillate, especially when the sample quantity is small. In contrast, the precision change curves of the Dynamic and PASTIS datasets with more categories are relatively stable. From the perspective of models, traditional model-driven AL algorithms have difficulty handling complex and diverse temporal image features. However, our method is data-driven and incorporates the concept of sample representativeness, resulting in more stable performance on these datasets.

Figure 8 illustrates the obtained classification maps for on the MUDS dataset. These classification maps reflect the prediction of the distribution of construction sites in a region of the United States. Entropy and Core-Set produced the most classification errors in non-building areas. The Random, Margin, and VAAL methods had fewer classification errors but still produced relatively ambiguous classification maps. The classification map of NNAL, which considers the characteristics of time series, is better in terms of visual results, with some noise around the non-building areas. However, our method achieved results that were closer to the ground truth.

Figure 9 illustrates classification maps for on the DynamicEarthNet dataset. These classification maps are predictions for features such as soil, forests, and agricultural areas in a Central American region. Random and VAAL obtained poor classification maps. Entropy and Margin are both uncertainty-based methods, and they are prone to selecting similar samples and falling into local optima. As can be discerned from the figure, the classification map generated by the Margin method encompasses a greater number of surface features belonging to the soil category. In contrast, the classification map yielded by the Entropy method incorporates a larger quantity of surface features falling within the agriculture category. Core-Set and NNAL consider the distances among samples when selecting samples. They chose more diverse samples, thus obtaining better classification maps, although there was still some noise present. Our method has obtained a relatively clear classification map, which is closer to the real classification result.

Figure 10 illustrates classification maps on the PASTIS dataset. These classification maps are predictions of the distribution of crops in a region of France. Random and VAAL obtained the most blurred classification maps. The two methods, Core-set and NNAL, also had limited performance when there were more data bands and a longer time series, resulting in classification maps with some noise. The uncertainty-based methods, Margin and Entropy, showed more obvious advantages when the number of classification categories in the dataset increased. Although there were cases of misclassification of winter rapeseed and winter barley, they had clearer classification maps. Our method selected and labeled samples based on classification loss, achieving a better classification effect.

### 4.6. Computational Cost

We carried out a study on the total computational cost of the proposed approach as well as the comparison methods during five rounds for the three given datasets. Table 3 presents training and sampling time in seconds. For training time, the four methods, namely, Random, Entropy, Margin, and Core-Set, have almost the same training time because the training process of these methods only involves the training of the classifier and does not involve the training of other network structures. Our method’s training phase consists of training the temporal classifier and the TLPM module, which takes more time than the methods mentioned above. VAAL requires training in the temporal classifier, VAE, and discriminator, and it takes a long time. For sampling time, Core-Set and NNAL will calculate the distances between samples at this stage, so they take more time than the baseline methods. Our method requires sufficient time to complete sample clustering and the prediction of the TLPM module.

## 5. Discussion

### 5.1. Experiments with Different Initial Training Sets

Figure 11 illustrates the comparative results of our method under varying initial training set sizes on three datasets, where every initial training set consists of randomly selected samples. We conducted repeated experiments using different initial training set sizes on the MUDS dataset. Our method outperforms other competitive methods across three different initial training set sizes. Experiments on the Dynamic dataset and the PASTIS dataset also indicate the exceptional performance of our method. The proposed method integrates representativeness and uncertainty. When the initial training set is small, our method will select representative time series samples to enrich the training set. When the size of the initial training set is relatively large, our method can automatically learn the distribution and characteristics of data from a large time series of data and select samples with high uncertainty. Therefore, our method can be applied to different sizes of the initial training set.

### 5.2. Ablation Study

Figure 12 illustrates our ablation study, which was conducted to investigate the contributions of the key modules, K-shape and self-attention, in our model. We conducted experiments on three datasets. Specifically, for the MUDS dataset, we set the initial training set sample count to 500 and selected a selection stride of 500. Similarly, for the DynamicEarthNet dataset and the PASTIS dataset, the initial training set sample count was set to 2000, with a selection stride of 1500. We performed four sets of experiments: (1) TLPM without self-attention, (2)TLPM + K-shape, (3) TLPM (with self-attention), and (4) TLPM (with self-attention) + K-shape (our method). Group (1) exhibited the lowest performance among the comparative methods. Group (3) outperformed Group (1), indicating that the TLPM without self-attention did not focus on the temporal characteristics of the time series samples when predicting loss, whereas the inclusion of self-attention enhanced the model’s predictive loss capability. Group (2) also showed good performance compared to Group (1) due to the consideration of sample distribution properties by the K-shape method. Notably, the model that incorporated both K-shape and self-attention demonstrated the highest performance, showcasing a significant advantage resulting from the simultaneous consideration of representativeness and uncertainty.

### 5.3. Experiments with Different Temporal Lengths

To explore the impact of different temporal lengths on classification accuracy, we conducted more experiments. For the MUDS dataset, 70% of the data is taken as the test set, 10% as the validation set, and 20% as the training set. For the DynamicEarthNet dataset and the PASTIS dataset, the division ratios of the test set, the validation set, and the training set are the same as those of the MUDS dataset. In the training set, we randomly cropped the data to lengths of 12, 16, 20,24, or 30, and then trained the classifier and tested the overall accuracy on the test set. Table 4 shows the relationship between the length of the time series and the overall accuracy. Obviously, an increase in the length of the time series can provide the model with more abundant contextual information and long-term dependency relationships. Therefore, when the time series is more complete, the overall accuracy is higher.

## 6. Conclusions

We have proposed an AL framework based on uncertainty and representativeness which has demonstrated obvious advantages in addressing the problem of sample scarcity in TSRSI classification. Our method introduces a data-driven mode, which can fully capture the temporal dependencies to find the key time series samples. Compared with the model-driven methods based on handcrafted features, it has better generalization performance. Our method has been verified on three datasets covering hundreds of locations around the globe. The experimental results prove that our method can be applied in a variety of different geographical environments. Experiments with different initial training sets indicate that our method is effective across different initial training set sizes. Ablation experiments were conducted, and the results confirmed the effectiveness of the key modules, K-shape, and self-attention. Meanwhile, our method also has certain limitations. For example, the sampling time is longer compared with other AL methods. We will further optimize the proposed method and its program code to accelerate its sampling speed in future research. In practical applications, our method enables the classifier to achieve a faster improvement in accuracy in the TSRSI classification tasks, which can be applied to scenarios such as crop monitoring [67], forest resource monitoring [68], water environment monitoring [69], etc.

## Figures and Tables

**Figure 1 sensors-25-01718-f001:**
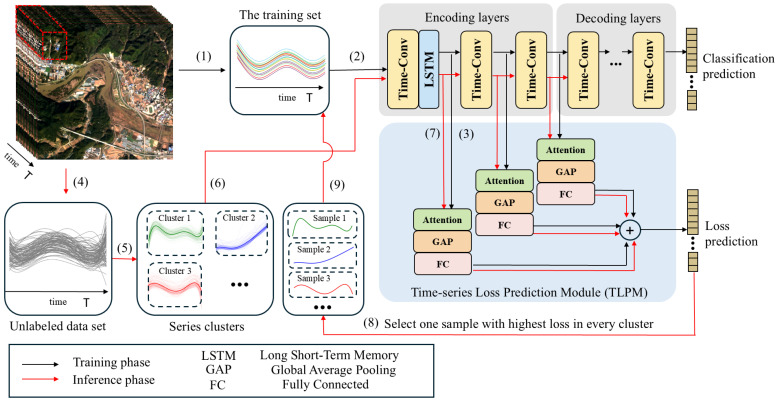
Flowchart of the proposed method. The main flow is divided into two phases: 1. Training phase: this phase involves training the temporal classifier and the Time-series Loss Prediction Module (TLPM). 2. Inference phase: this phase includes the clustering process, loss prediction, and sample selection.

**Figure 2 sensors-25-01718-f002:**
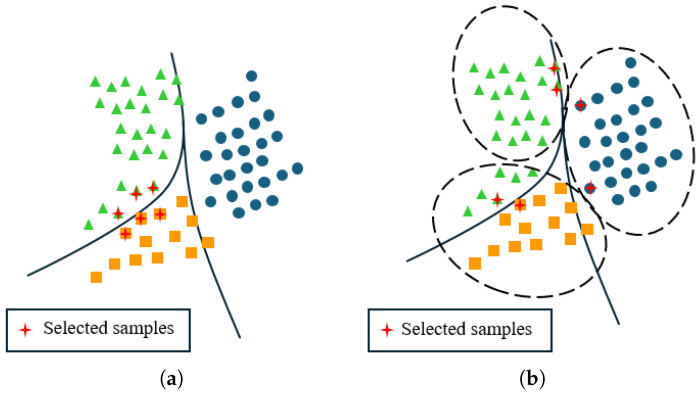
The situations of samples that different AL methods tend to select. (**a**) The AL method based on uncertainty. (**b**) The AL method based on both representativeness and uncertainty.

**Figure 3 sensors-25-01718-f003:**
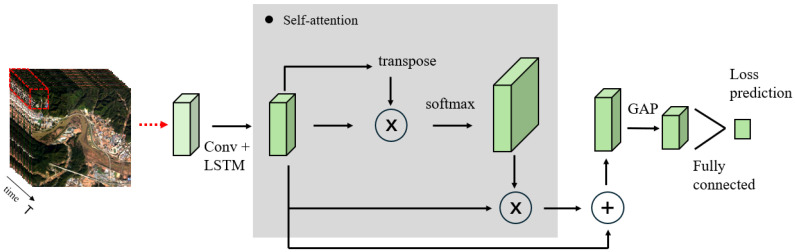
One of the network branches of the Time-series Loss Prediction Module (TLPM). This network branch is composed of a self-attention mechanism, a Global Average Pooling (GAP) layer, and a fully connected (FC) layer. The complete TLPM can be seen in Figure 1.

**Figure 4 sensors-25-01718-f004:**
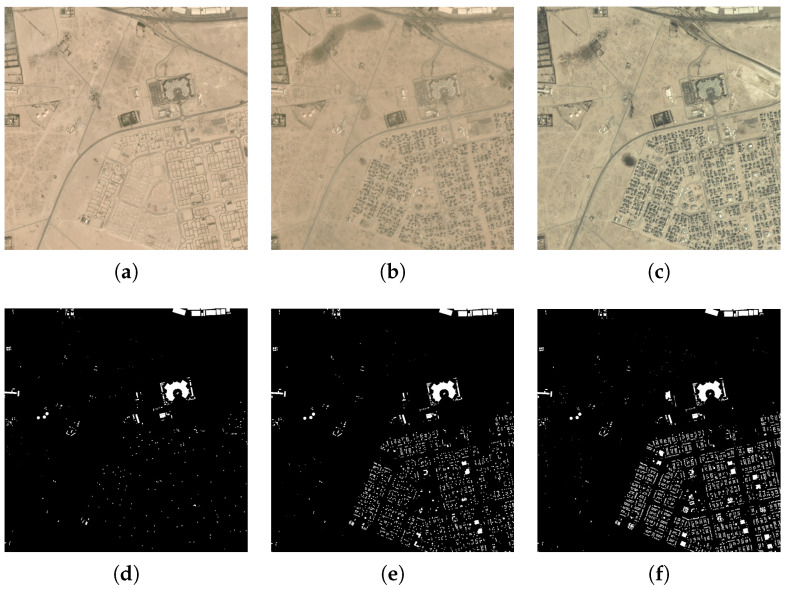
The sample images of MUDS and their ground-truth classification maps. They are taken from a region of the United States. In their ground-truth classification map, the white areas represent the built-up areas, while the black areas represent the non-built-up areas. We can clearly observe that the construction areas are gradually increasing. (**a**) An original image of the MUDS dataset in January 2018. (**b**) An original image of the MUDS dataset in January 2019. (**c**) An original image of the MUDS dataset in January 2020. (**d**) The ground-truth classification map from the MUDS dataset in January 2018. (**e**) The ground-truth classification map from the MUDS dataset in January 2019. (**f**) The ground-truth classification map from the MUDS dataset in January 2020.

**Figure 5 sensors-25-01718-f005:**
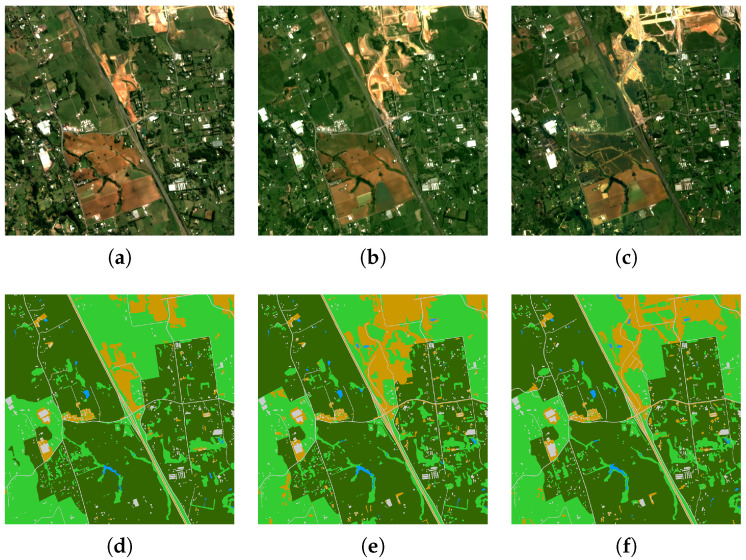
The sample images of DynamicEarthNet and their ground-truth classification maps. These images were taken from a region in New Zealand, which include land use types such as soil, forest, agricultural land, etc. In the figure, we can see that the forest area initially decreased and then increased. (**a**) An RGB image of the DynamicEarthNet dataset in January 2018. (**b**) An RGB image of the DynamicEarthNet dataset in December 2018. (**c**) An RGB image of the DynamicEarthNet dataset in December 2019. (**d**) The ground-truth classification map from the DynamicEarthNet dataset in January 2018. (**e**) The ground-truth classification map from the DynamicEarthNet dataset in December 2018. (**f**) The ground-truth classification map from the DynamicEarthNet dataset in December 2019.

**Figure 6 sensors-25-01718-f006:**
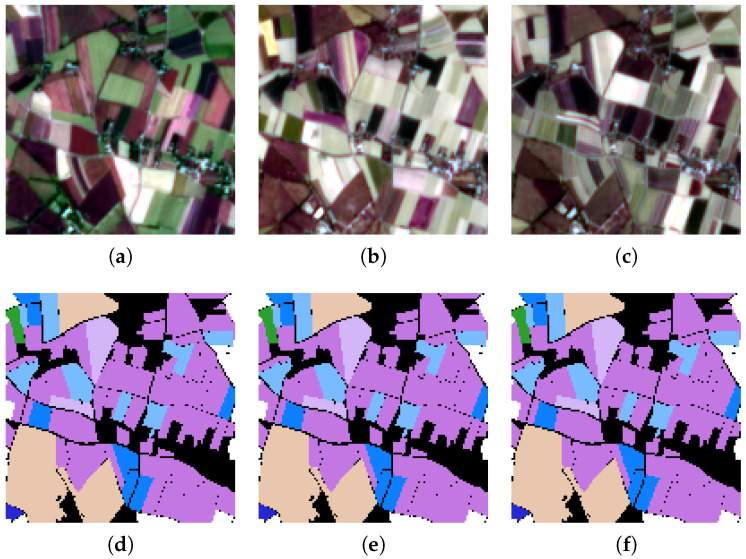
The sample images of PASTIS and their ground-truth classification maps. They were taken from a region of France and reflect crop distributions, with a spatial resolution of 10 m. (**a**) The RGB image from the PASTIS dataset in February 2019. (**b**) The RGB image from the PASTIS dataset in July 2019. (**c**) The RGB image from the PASTIS dataset in August 2019. (**d**) The ground-truth classification map from the PASTIS dataset in February 2019. (**e**) The ground-truth classification map from the PASTIS dataset in July 2019. (**f**) The ground-truth classification map from the PASTIS dataset in August 2019.

**Figure 7 sensors-25-01718-f007:**
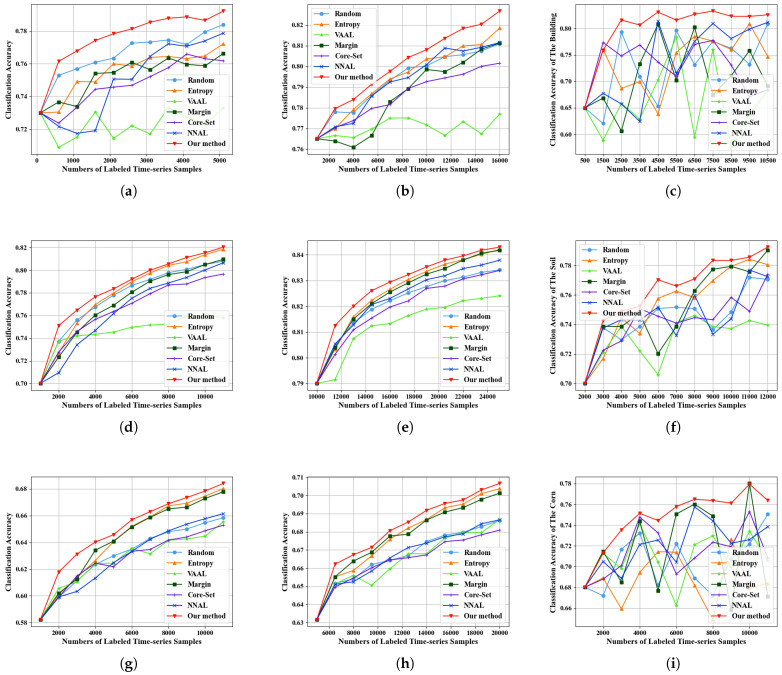
The experiment results on three datasets. (**a**) The experiment results in OA with an initial sample size of 100 and a sample selection increment of 500 on MUDS. (**b**) The experiment results in OA with an initial sample size of 1000 and a sample selection increment of 1500 on MUDS. (**c**) The classification accuracy of the building on MUDS. (**d**) The experiment results in OA with an initial sample size of 1000 and a sample selection increment of 1000 on DynamicEarthNet. (**e**) The experiment results in OA with an initial sample size of 10,000 and a sample selection increment of 1500 on DynamicEarthNet. (**f**) The classification accuracy of the Soil on DynamicEarthNet. (**g**) The experiment results in OA with an initial sample size of 1000 and a sample selection increment of 1000 on PASTIS. (**h**) The experiment results in OA with an initial sample size of 5000 and a sample selection increment of 1500 on PASTIS. (**i**) The classification accuracy of the Corn on PASTIS.

**Figure 8 sensors-25-01718-f008:**
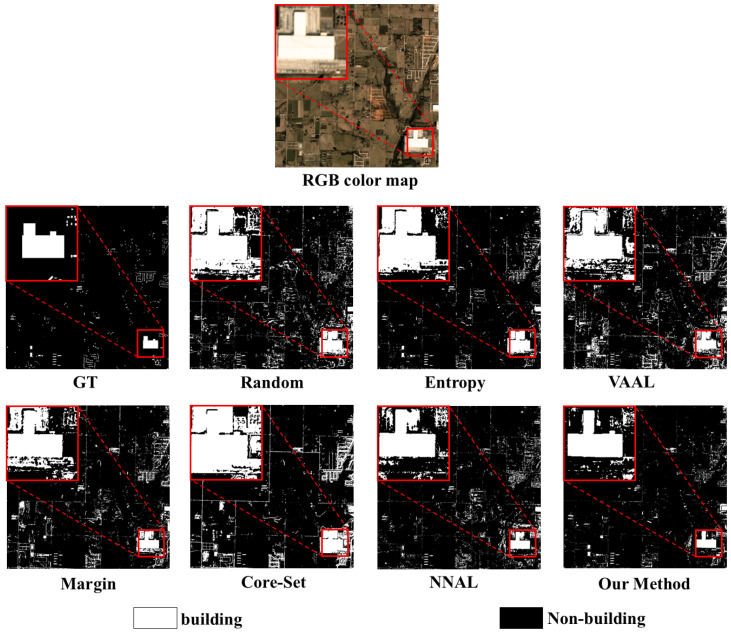
Classification map comparisons of the compared methods for 16,000 labeled samples in the MUDS dataset.

**Figure 9 sensors-25-01718-f009:**
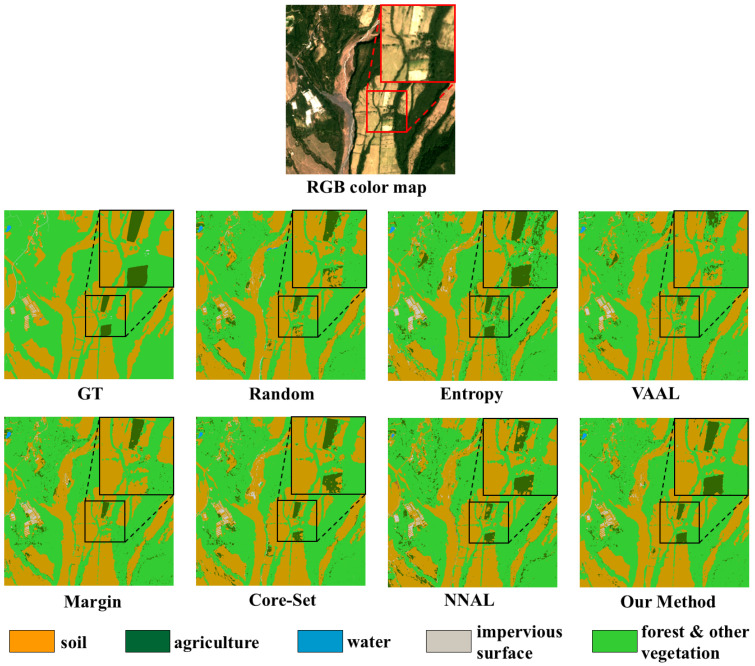
Classification map comparisons of the compared methods for 17,500 labeled samples in the DynamicEarthNet dataset.

**Figure 10 sensors-25-01718-f010:**
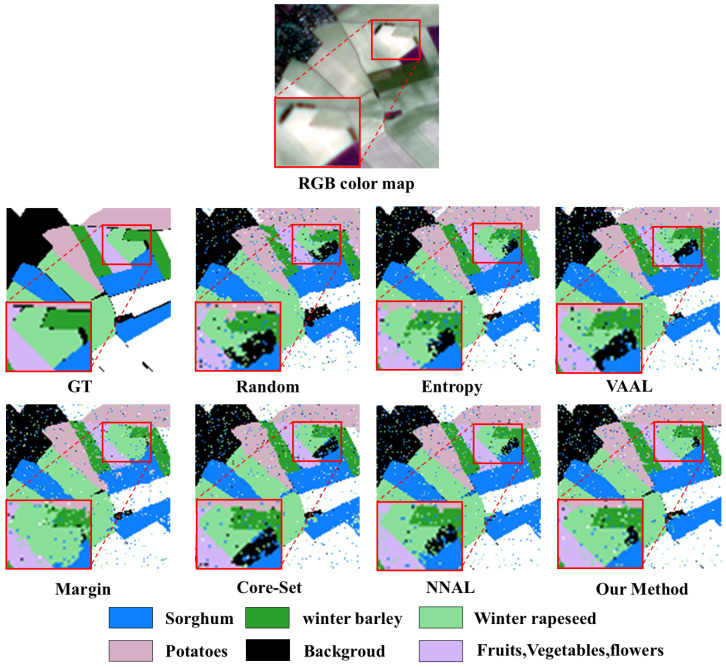
Classification map comparisons of the compared methods for 40,000 labeled samples in the PASTIS dataset.

**Figure 11 sensors-25-01718-f011:**
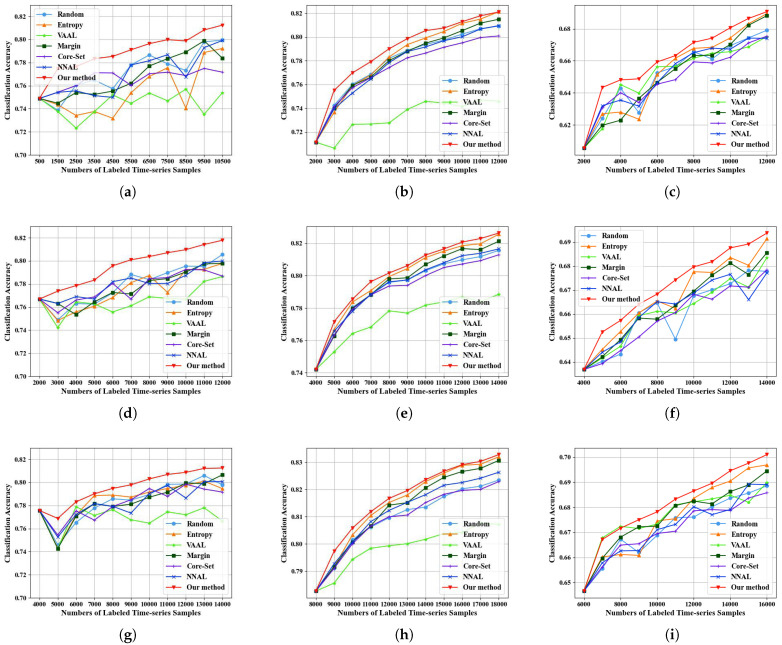
The experiment results with different initial training sets. (**a**) The experiment results with an initial sample size of 500 and a sample selection increment of 1000 on MUDS. (**b**) The experiment results with an initial sample size of 2000 and a sample selection increment of 1000 on DynamicEarthNet. (**c**) The experiment results with an initial sample size of 2000 and a sample selection increment of 1000 on PASTIS. (**d**) The experiment results with an initial sample size of 2000 and a sample selection increment of 1000 on MUDS. (**e**) The experiment results with an initial sample size of 4000 and a sample selection increment of 1000 on DynamicEarthNet. (**f**) The experiment results with an initial sample size of 4000 and a sample selection increment of 1000 on PASTIS. (**g**) The experiment results with an initial sample size of 4000 and a sample selection increment of 1000 on MUDS. (**h**) The experiment results with an initial sample size of 8000 and a sample selection increment of 1000 on DynamicEarthNet. (**i**) The experiment results with an initial sample size of 6000 and a sample selection increment of 1000 on PASTIS.

**Figure 12 sensors-25-01718-f012:**
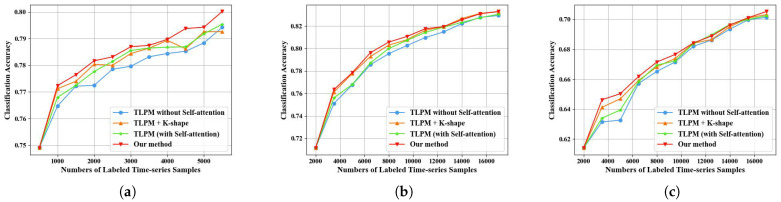
Ablation results on analyzing the effect of the K-shape and self-attention. (**a**) The experiment results with an initial sample size of 500 and a sample selection increment of 500 on MUDS. (**b**) The experiment results with an initial sample size of 2000 and a sample selection increment of 1500 on DynamicEarthNet. (**c**) The experiment results with an initial sample size of 2000 and a sample selection increment of 1500 on PASTIS.

**Table 1 sensors-25-01718-t001:** Dataset details.

Dataset	DynamicEarthNet	MUDS	PASTIS
Resolution (m)	3	4	10
Sensor	Planet Labs	Planet Labs	Sentinel-2
Bands	4	4	10
Temporal length	2 years	2 years	more than 2 years
Sample frequency	monthly	monthly	irregular
Categories	7	2	19

**Table 2 sensors-25-01718-t002:** Kappa coefficients of the comparative methods. In this table, the MUDS dataset has 16,000 labeled samples, the Dynamic dataset has 24,000 labeled samples, and the PASTIS dataset has 20,000 labeled samples.

Dataset	MUDS	DynamicEarthNet	PASTIS
Random	0.5897	0.7440	0.6022
Entropy	0.6257	0.7731	0.6119
VAAL	0.5139	0.6949	0.5937
Margin	0.5856	0.7614	0.6041
Core-set	0.5590	0.7470	0.5887
NNAL	0.5539	0.7595	0.5641
Our method	0.6663	0.7845	0.6231

**Table 3 sensors-25-01718-t003:** Training and sampling time in seconds of the comparison methods.

AL Methods	MUDS		DynamicEarthNet		PASTIS	
	Training	Sampling	Training	Sampling	Training	Sampling
Random	123.25	0.28	179.21	0.78	198.95	0.72
Entropy	125.15	3.28	183.73	4.13	194.44	6.81
VAAL	285.69	8.70	450.84	9.73	682.36	11.43
Margin	125.75	2.63	180.86	4.22	195.23	5.98
Core-set	129.50	31.26	178.25	42.02	192.73	245.33
NNAL	126.05	124.25	181.27	83.67	197.92	215.09
Our	197.49	315.46	223.47	342.96	281.83	387.13

**Table 4 sensors-25-01718-t004:** The relationship between the length of the time series and the overall accuracy.

Temporal Length	DynamicEarthNet	MUDS	PASTIS
12	79.37%	79.34%	70.80%
16	80.86%	79.96%	71.64%
20	81.96%	81.35%	71.82%
24	84.61%	84.78%	72.57%
30	-	-	74.02%
36	-	-	75.54%

## Data Availability

The DynamicEarthNet dataset used in this paper is available at https://mediatum.ub.tum.de/1650201 (accessed on 30 May 2024). The MUDS dataset used in this paper is available at https://registry.opendata.aws/spacenet/ (accessed on 30 May 2024). The PASTIS dataset used in this paper is available at https://zenodo.org/records/5012942 (accessed on 30 January 2025). The datasets generated or analyzed during this study are available from the corresponding author upon reasonable request. Our code is available at https://github.com/Fighting-Golion/time_series_active_learning (accessed on 2 March 2025).

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
