# Peer review of "Time Series Remote Sensing Image Classification with a Data-Driven Active Deep Learning Approach"

_sensors, 2025, doi:10.3390/s25061718_

Round 1
Reviewer 1 Report
Comments and Suggestions for Authors
The manuscript reports a deep learning approach for time series image classification. The text is prepared well. My concerns are as following.
1. Image classification is an application-pushed work. The accuracy may depend on how many classes and which class. The classes included and the definition of each class is the first thing. Then the suitable method is selected. However, in this manuscript, the name and definition of classes are not described clearly. One case includes two classes. Another includes seven. Both far less than widely used land cover and land use datasets. If the classes just very few classes, for instance, the built-up area and non-built-up area, I think the accuracy of many other approach is also acceptable. Then what is the privilege of the method introduced in this manuscript?
2. For the circumstances of many classes in complex environments, more data sources may be critical. The accuracy depends on the data sources and the geographical region selected. This paper tries to illustrate the effectiveness of the methods. So, the first thing is to avoid the impacts of data sources and geographical regions. I suggest the author clearly describe the suitability of this classification approach. From this paper, it is clear the approach is suitable for sampled datasets of DynamicEarthNet and MUDs. However, I am not sure this methods works for all DynamicEarthNet and MUDs datasets over the world, not mention other data sources. The suitability of the approach should be test in geographical regions with differed landscape complexity, and for more data sources.
Author Response
Thank you very much for taking the time to review this manuscript. Please find our point-by-point responses in the PDF attachment. The re-uploaded highlighted version demonstrates the revisions we have made to the paper.

Reviewer 2 Report
Comments and Suggestions for Authors
It is necessary to include statistical values ​​of the results in the abstract, as well as the software used for comparison with traditional supervised and unsupervised classification methods, as well as the Kappa indices of the images used.

Author Response

(The authors gave the same response as above.)

Reviewer 3 Report
Comments and Suggestions for Authors
Introduction
We appreciate the didactic explanation of the study basic ideas. Perhaps you can shorten them a little and bring one or two examples of geographical contexts of the application of DL with AL expert based knowledge. For example mountain terrain imagery can be a situation where complex spectral signature of forest stands changes in time...
The innovation components listed can be better turned and adapted as objectives of the approach.
Related work section seems to be a part of the methodological part (first sub-section).
Methodology
Figure 1 is a part of methodology as a core element of the study. It can be moved there. Explain the acronyms in figure explanation as well.
Mathematical approach can be more simple in order to be easily understood for all potential readers.
Results
We recommend to try finding an equal proportion between the formulae section and the exemplification with satellite data. For example it is better to start the experimental approach with the basic formulae (explained) and the connection with the image specific features to be classified. The uncertainty component can be also explained on the selected case study.
Table 1 and figures 4-5 need more explanation.
The spatial and temporal context of the study selected is useful in opening the paper to other readers like Earth scientists. Some terrain features and phenomenological approaches needs a little more explanation. It is essential to link the sensor capacity to detect object in imagery and the accuracy and even the precision of the experimental results based on selected samples like building(s), roads, forest stands etc.
The goal is to produce competitive classification results starting from a limited effort in AL expert based contribution. In this respect it can be really interesting to provide some examples. These are also presented in figures 7 and 8 (needs more interpretation).
Discussion
The results are calibrated from the statistical point of view. Our interest is to provide some qualitative significance of the classified data and in relationship to a specific phenomenon. The second aspect is the explanation of the relationship between the proposed method and the image features like resolution (including radiometric) and their pluses and minuses, including the geometric errors and of course the radiometric correction imprints (if it does exist).
If talking on time series is it an influence of the size of the image cube on the results?
Conclusion
Some practical or applied aspects can be emphasized. Where the experimental results can be applied?
Author Response

(The authors gave the same response as above.)
